# Butyric Acid from Probiotic *Staphylococcus epidermidis* in the Skin Microbiome Down-Regulates the Ultraviolet-Induced Pro-Inflammatory IL-6 Cytokine via Short-Chain Fatty Acid Receptor

**DOI:** 10.3390/ijms20184477

**Published:** 2019-09-11

**Authors:** Sunita Keshari, Arun Balasubramaniam, Binderiya Myagmardoloonjin, Deron Raymond Herr, Indira Putri Negari, Chun-Ming Huang

**Affiliations:** 1Department of Life Sciences, National Central University, Taoyuan 32001, Taiwan; 2Department of Biomedical Sciences and Engineering, National Central University, Taoyuan 32001, Taiwan; 3Department of Pharmacology, National University of Singapore, Singapore 117600, Singapore; 4Department of Dermatology, School of Medicine, University of California, San Diego, CA 92093, USA

**Keywords:** butyric acid, microbiome, probiotic, *S. epidermidis*, UVB

## Abstract

The glycerol fermentation of probiotic *Staphylococcus epidermidis* (*S. epidermidis*) in the skin microbiome produced butyric acid in vitro at concentrations in the millimolar range. The exposure of dorsal skin of mice to ultraviolet B (UVB) light provoked a significant increased production of pro-inflammatory interleukin (IL)-6 cytokine. Topical application of butyric acid alone or *S. epidermidis* with glycerol remarkably ameliorated the UVB-induced IL-6 production. In vivo knockdown of short-chain fatty acid receptor 2 (FFAR2) in mouse skin considerably blocked the probiotic effect of *S. epidermidis* on suppression of UVB-induced IL-6 production. These results demonstrate that butyric acid in the metabolites of fermenting skin probiotic bacteria mediates FFAR2 to modulate the production of pro-inflammatory cytokines induced by UVB.

## 1. Introduction

Skin is a fundamental component of the innate immune system, providing a protective barrier against the penetration of microorganisms and preventing environmental damage to the body. Exposure to UV is known to induce clustering of specific cell-surface receptors, resulting in the activation of signal transduction pathways [1]. Prolonged exposure of ultraviolet B (UVB) with a wavelength of 280–320 nm is a significant risk factor causing chronic inflammation, DNA damage, and lipid peroxidation in the epidermis [2]. Chronic exposure to this cellular stressor commonly results in skin pathologies such as epidermal hyperplasia, erythema, and edema [3]. Within the skin epidermis, keratinocytes have been found to play a critical role in the response to photo-damage to UVB by releasing key inflammatory mediators such as interleukin (IL)-1, -6, -8, -10 and tumor necrosis factor (TNF). Among these, keratinocyte-derived IL-1 and IL-6 seem to be of particular importance [4]. Many studies have demonstrated that both human and murine keratinocytes produce these cytokines because of UV exposure [5,6]. Continuous exposure to UV results in the generation of reactive oxygen species (ROS), exposing a variety of biomolecules to oxidative stress. This leads to damage of biological structures, contributes to cellular injury, and ultimately resulting in tissue destruction [7]. 

The skin microbiome consists of a complex community of organisms that mediates essential physiological and pathological processes. Among these organisms, *Staphylococcus epidermidis* (*S. epidermidis*) is the predominant commensal bacterial species that colonizes normal human skin [8]. Derangement of the bacterial milieu can alter epidermal function, and thus, disrupt the local ecology of the skin [9]. This interplay of microbe and host appears to be critical for establishing homeostasis. *S. epidermidis* has also been shown to benefit the skin immune function by diminishing inflammation after injury [10]. Study have shown that 6-*N*-hydroxyaminopurine, a metabolite from *S. epidermidis* could selectively inhibit UV induced skin tumors in mice [11]. Additionally, studies have provided evidence that peptides and other metabolites produced from *S. epidermidis*, might also act as antimicrobial agents and contribute to normal defense at the epidermal interface [12,13]. In the current study, we demonstrate that butyric acid, one of the major metabolites produced by *S. epidermidis* fermentation, attenuates the UVB-induced production of inflammatory cytokines by interacting with cognate receptors expressed by keratinocytes. This suggests that manipulation of the *S. epidermidis*-butyric acid-FFAR2 axis is a potential therapeutic target to protect against skin inflammatory diseases. 

## 2. Results

### 2.1. S. epidermidis Mediates Glycerol Fermentation and Acetolactate Synthase (ALS) Activity

To investigate whether *S. epidermidis* (ATCC 12228) can ferment glycerol, *S. epidermidis* (10^7^ colony-forming unit (CFU) were incubated with and without glycerol (2%) in rich media with phenol red for 12 h. Uninoculated rich media containing phenol red with glycerol served as controls. Phenol red was used as an indicator to monitor bacterial fermentation. In rich media incubated with bacteria, the color of phenol red changed from red to orange because of the bacterial replication during incubation. However, in agreement with earlier results, the rich media containing bacteria along with glycerol, the phenol red color changed from red to yellow indicating the use of glycerol as a carbon source for fermentation by *S. epidermidis* after 12 h culture (Figure 1a) [13,14,15]. The color change of phenol red was quantified by measuring the optical density at 560 nm (OD_560_) (Figure 1b). Furthermore, OD_560nm_ in rich media with *S. epidermidis* and glycerol was significantly lower than that in rich media with *S. epidermidis* alone (Figure 1a). Next, to confirm the fermentation activity of *S. epidermidis* using glycerol as a carbon source, we added furfural, a potent fermentation inhibitor [16], in rich media containing *S. epidermidis* and glycerol. Media with and without *S. epidermidis*, glycerol and furfural were used as controls. We did not detect any color change in the media incubated with *S. epidermidis* with furfural in the presence or absence of glycerol (Figure 1a). Additionally, the OD_560nm_ was comparable to the control group containing media only, or media with furfural, or media with glycerol. This indicates that the fermentation or metabolic activity of *S. epidermidis* might have been inhibited by furfural (Figure 1b). Microbial enzymes are found to be the major source of bacterial fermentation by catalyzing the hydrolysis of starch or peptides. Moreover, a recent study reported that acetolactate synthase (ALS) is a crucial enzyme in *Staphylococcus aureus* (*S. aureus*) which converts pyruvate into α-acetolactate, promoting the production of branched-chain amino acids (BCAA) and playing an important role in the activation of the tricarboxylic acid (TCA) cycle [17]. It also promotes anaerobic fermentation in *Bacillius subtilis* (*B. subtilis*) by converting two molecules of pyruvate to form acetolactate, functioning as a key regulatory enzyme in fermentation [18]. Reports also demonstrated that furfural exerts its inhibitory effects on microbial fermentation via interfering with the activities of enzymes in the pathways of microbial fermentation [19]. We confirmed that the ALS enzyme activity of *S. epidermidis* is significantly reduced when it is incubated with furfural (inhibitor) (Figure 1c). Taken together, we confirmed that *S. epidermidis* induced glycerol fermentation, which required furfural-sensitive ALS activity.

### 2.2. Butyric Acid, a Product in Fermented Media upon Glycerol Fermentation by S. epidermidis

Previous studies have detected the presence of multiple short-chain fatty acids (SCFAs) such as acetic acid, butyric acid, lactic acid, and succinic in fermented media of *S. epidermidis* from glycerol fermentation [13]. Butyric acid from *S. epidermidis* was found to exert growth suppressive effects on USA300, a community-associated methicillin-resistant *S. aureus* (MRSA). Moreover, it shows potent anti-inflammatory activity in skin keratinocytes by effectively inhibiting histone deacetylase (HDAC) [20,21]. In the current study, we have screened the supernatant following glycerol fermentation of *S. epidermidis* to quantify their butyric acid producing capacity by high-performance liquid chromatography-ultraviolet (HPLC-UV) analysis. Butyric acid was detected as a sharp specific peak in the HPLC chromatogram and was determined to be at a concentration of 6 mM in the fermented media by comparison to a butyric acid standard curve (Figure 2a,b).

### 2.3. Mixture of S. epidermidis and Glycerol or Butyric Acid alone Inhibited UVB Induced Skin Inflammation

Exposure to prolonged UVB radiation is deleterious to human skin, which may lead to expression and secretion of pro-inflammatory cytokines such as IL-6 in human keratinocytes [22]. Furthermore, chronic UVB is responsible for epidermal hyperplasia. The dorsal skin of the Institute of Cancer Research (ICR) mice were shaved and exposed to 12 doses of UVB (195 mJ/cm^2^) over 4 weeks concurrent with 4 topical applications of a mixture of *S. epidermidis* (10^7^) and glycerol (2%) in phosphate-buffered saline (PBS) over the same time period. Mice topically applied with mixture of *S. epidermidis* (10^7^) and glycerol (2%) in PBS with no UVB exposure were included as control. The dorsal skin of mice was photographed to measure the wound healing from 0, 16, and 30 days. Mice from all groups were sacrificed on the 30th day and skin samples were lysed to extract total protein. In mice treated with either PBS, *S. epidermidis*, or glycerol alone, variable lesions were recorded which started from edema, redness, and ulceration of the UVB exposed area and ended with thickening of the skin. However, in mice that received topical applications of *S. epidermidis* and glycerol, a comparatively lower level of erythema and ulceration were observed on UVB-exposed skin on day 16. Further improvement was noticed on day 30, signifying maximum recovery from UVB damage (Figure 3a,b) compared to non-exposed groups (Appendix A). Upregulated levels of IL-6 and noticeable epidermal hyperplasia (as quantified by epidermal thickness) were detected in mice skin topically applied with PBS, *S. epidermidis*, or glycerol alone following UVB exposure because of an increase in the number of keratinocytes and epidermal layers. By contrast, in the *S. epidermidis* plus glycerol treated group, IL-6 levels were significantly reduced (Figure 3c) compared to non-exposed groups (Appendix A) and the skin remained almost completely intact, showing a marked attenuation of UVB-induced skin thickening (Appendix A). This indicates that metabolites from glycerol fermentation by *S. epidermidis* might induce anti-inflammatory activity against chronic UVB radiation. To further confirm the anti-inflammatory activity of *S. epidermidis* fermentation, another group of mice were topically applied with a mixture of *S. epidermidis* and glycerol along with fermentation inhibitor, furfural. UVB irradiation-induced skin lesions, elevation of IL-6, and epidermal hyperplasia were not ameliorated upon topical application of *S. epidermidis* plus glycerol and furfural compared to control group receiving water and furfural alone (Figure 3d and Appendix A). However, significant recovery from inflammation was detected in mice topically applied with a mixture of *S. epidermidis* and glycerol. This was again accompanied by a downregulated level of IL-6 and decreased epidermal thickness, confirming that glycerol fermentation from *S. epidermidis* exerts anti-inflammatory activity. In the present study, we detected the production of butyric acid in significant amount as a metabolite in the fermentation of glycerol by *S. epidermidis*. Since butyric acid has been shown to be a potent anti-inflammatory agent in skin keratinocytes and has been tested for the attenuation of different inflammatory disorders [23,24], we first investigated its role against UVB induced inflammation in human keratinocytes (CCD 1106 KERTr). The human keratinocytes were irradiated with 195 mJ/cm^2^ UVB and the IL-6 production was dramatically enhanced in water control group until 12 h after irradiation. However, the addition of BA (butyric acid, 4 mM), significantly reduced the upregulated level of IL-6 (Appendix A). As studies mentioned, both IL-6 and IL-8 undoubtedly play pivotal roles in immunologic regulation in human skin and are involved in skin inflammation in response to UV radiation [25]. We have also detected the upregulated level of IL-8 in UVB exposed keratinocytes, which further rescued upon BA treatment (Appendix A). Further, we detected effect of BA application upon UVB exposure in vivo. Our results show that topical application of 4 mM butyric acid ameliorated UVB-induced wound formation, and wound healing was mild to significant from day 16 to day 30 (Appendix A). In a similar manner, IL-6 was downregulated and epidermal hyperplasia was diminished (Figure 3e and Appendix A), which was consistent with the model that butyric acid is the active glycerol fermentation metabolite by which *S. epidermidis* can inhibit UV induced inflammation in the skin.

### 2.4. Knocking Down Free Fatty Acid Receptor 2 (FFAR2) Inhibited Butyric Acid Mediated Attenuation of Inflammation in Chronic UVB

Most beneficial roles of SCFAs in the gastrointestinal tract (GI) are mediated by directly activating its cognate G protein-coupled receptor, GPR43 (also known as FFAR2). For example, the chemoattractant and anti-inflammatory activities of butyrate in chondrocytes are completely mediated by its binding to GPR43 [26]. The activation of FFAR2 has emerged as a potentially important mechanism by which SCFAs could directly regulate the immune cells and the process of inflammatory diseases, thus drawing much attention in recent years [27,28]. Considering the key function of FFAR2 in inflammatory processes, blocking the signaling of this receptor by a selective inhibitor could be a useful approach to evaluate the role of butyric acid in the regulation of immune function in skin during chronic UVB exposure. In the present study, FFAR2 was inhibited by via gavage feeding of FFAR2 selective antagonist GLPG0974 (0.1–1 mg/kg) 20 min prior to topical application of butyric acid followed by UVB exposure. In addition, we have also blocked FFAR2 receptor via subcutaneous injection of FFAR2 siRNA (5 µM) into the dorsal skin of ICR mice 20 min prior to topical application of butyric acid followed by UVB exposure. In both conditions of blocking FFAR2, significant wound area and redness from UVB radiation was observed (Appendix A) on mice dorsal skin, which further did not recover even after topical application of butyric acid. Inflammation, as detected by upregulation of IL-6 and epidermal hyperplasia, was increased in mice upon FFAR2 inhibition and further remained unchanged even on topical application of butyric acid compared to mice applied with butyric acid alone (Figure 4a,b and Appendix A). We have also confirmed the FFAR2 inhibition or gene knockdown by measuring the protein expression level of FFAR2 by Western blot analysis and mRNA relative expression by RTPCR analysis (Appendix A). Our overall results demonstrate an essential role of FFAR2/GPR43 in butyrate activity in UVB-induced chronic inflammation process. Butyrate reduced pro-inflammatory mediators IL-6, by binding to its FFAR2 receptor, which further inhibits the epidermal hyperplasia in chronic UVB exposure.

## 3. Discussion

Skin and its constituent epidermal keratinocytes make-up the outer most protective covering of the body, and as such, are the primary targets for solar UVB radiation. Exposure to chronic UVB radiation alters the cutaneous and systemic immune systems by causing several pathological alteration in cell signaling resulting in sunburn, erythema, inflammation, and skin carcinogenesis [29]. It primarily activates various pro-inflammatory mediators such as tumor necrosis factor-alpha (TNF-α), cyclooxygenase-2 (COX-2), and IL-6, leading to the activation of nuclear factor kappa-light-chain-enhancer of activated B cells (NF-κB) [30]. The findings from the previous studies showed that *S. epidermidis*, a commensal bacteria from human skin, could ferment glycerol into butyric acid, one of the four SCFAs detected in fermented media [13,15]. Glycerol is a major component in stratum corneum (SC) hydration with varied amount in different parts of human skin (0.7 μg/cm^2^ in the cheek and 0.l μg/cm^2^ in forearm and sole). Multiple diseases characterized by xerosis and impaired epidermal barrier function, such as atopic dermatitis were improved by topical application of glycerol. Although glycerol could not penetrate deeper in SC but the SCFAs generated from glycerol fermentation by skin commensal bacteria, may modulate physiological processes such as skin barrier homeostasis and inflammation [15,31,32]. In this study, we evaluated the role of butyric acid, a major SCFA from the fermentation of glycerol by *S. epidermidis*, on chronic UVB radiation induced inflammation on ICR mice skin. The fermentation of glycerol by *S. epidermidis* was detected by change in color of phenol red from red to yellow (more acidic), which was further validated quantitatively by a significant decrease in OD_560nm_. In addition to SCFAs, furfural is another metabolite produced by bacterial fermentation, and its accumulation has been shown to inhibit fermentation by blocking the activities of multiple enzymes involved in bacterial fermentation pathway [19,33]. We did not detect fermentation by color change or by OD_560nm_ quantification upon adding furfural to media containing *S. epidermidis* with glycerol. Furthermore, no bacterial growth was detected when furfural was added to media containing *S. epidermidis* alone, demonstrating the activity of furfural in suppressing bacterial fermentation activity. ALS is one of the major fermentation enzymes catalyzing the conversion of pyruvate to α-acetolactate which is further converted to isobutyrate and then ultimately to butyrate in subsequent pathways [34,35]. Our recent study showed, ALS activity in *S. epidermidis* was inhibited by a furfural derivative 5-methyl furfural (5MF) [14]. In the current study, we detected a significant reduction in ALS activity upon incubation of bacterial lysate with furfural. This demonstrates that ALS plays an essential role in *S. epidermidis* metabolism, triggering glycerol fermentation. We detected the production of butyric acid in fermented media by HPLC analysis on the basis of comparing the retention time and spectra with a butyric acid standard, confirming butyric acid as one of the most abundant metabolites produced from fermentation of 2% glycerol by *S. epidermidis*.

Metabolites from microbial fermentation have been under investigation as potential therapeutics against different infectious and inflammatory diseases [36]. For example, research suggested that ingestion of probiotic bacteria, *Lactobacillus johnsonii* (La1) could protect against the UVR-induced inflammation [37]. Furthermore, several lines of evidence suggest that metabolites from the fermentation of glycerol by *S. epidermidis* can inhibit the growth of pathogenic *Cutibacterium acnes* (*C. acnes*), thus, *S. epidermidis* may be an excellent probiotic for the treatment of chronic sinusitis [13,15,38]. Here we detected the probiotic effect of *S. epidermidis* in skin against UVB mediated chronic inflammation. Topical application of *S. epidermidis* or glycerol alone on ICR mice dorsal skin exposed to UVB showed noticeable erythema, ulceration, epidermal hyperplasia, and increased IL-6 levels. Following UVB exposure, increased infiltration of dermal inflammatory cells may lead to epidermal proliferation of keratinocytes and hyperplastic lesion [39,40]. A significant decrease in inflammation with attenuation of IL-6 and a reduction of epidermal thickness was observed in skin topically applied with mixture of *S. epidermidis* and glycerol, suggesting that the anti-inflammatory effect observed here is likely to be mediated through *S. epidermidis* fermentation using glycerol as a carbon source. Further, we confirmed that the anti-inflammatory activity was the result of *S. epidermidis* fermentation with the use of the fermentation inhibitor, furfural. Control mice topically applied with water or furfural alone followed by irradiation with UVB appeared to have significant ulceration, epidermal thickening, and increased IL-6. Importantly, there was no recovery from ulceration or inflammation in skin applied with a mixture of *S. epidermidis* and glycerol along with furfural. Based on these findings, we infer that a product from the fermentation of glycerol by *S. epidermidis* could inhibit inflammation from chronic UVB. Bacterial interference using live beneficial bacteria is a promising method of preventing or treating many infections. Because of the US Food and Drug Administration (FDA) restriction in use of live probiotic bacteria for cosmetics, the promotion of beneficial effects through the generation of fermentation metabolites from live probiotic bacteria has become a desirable alternative [28,29]. Consistent with a previous study [13], the current work used HPLC analysis to detect a substantial amount of butyric acid in the fermented media of *S. epidermidis* using glycerol as a carbon source. Anti-microbial, anti-inflammatory, and anti-cancer activity of butyric acid and its derivatives make it as a potential drug of choice [41,42]. It has been shown that application of both butyric acid and its co-drug could reportedly inhibit skin tumorigenesis and psoriasis-like skin inflammation [43,44]. In the current study, topical application of butyric acid markedly decreases skin ulceration and epidermal thickness from UVB exposure compared to the control mice skin applied with H_2_O. In line with the anti-inflammatory activity of butyric acid, a significant reduction in IL-6 level was detected both in keratinocytes and in mice skin exposed to prolonged UVB, indicating its potential use as an agent to prevent inflammation in chronic UVB exposed skin. Reduction in level of IL-8 upon butyric acid treatment, provided new insight for the application of butyric acid against other upregulated cytokines against UVB induced inflammation.

Research demonstrated that butyrate can control immune/inflammatory reactions in the skin [42]. SCFAs exert their immunomodulatory effects via binding to its cognate receptor, GPR43 [45]. Exacerbated or unresolved inflammation was observed in GPR43-deficient (Gpr43^−/−^) mice in models of colitis, arthritis, and asthma, demonstrating that SCFA-GPR43 interactions are obligatory for normal resolution of certain inflammatory responses [27]. Moreover, decreased expression of GPR43 by keratinocytes in psoriatic skin could be rescued by topical application of sodium butyrate, indicating that butyrate-GPR43 interactions might exert anti-inflammatory effects in psoriasis [46]. In our study, gavage feeding of mice with GLPG0974, an antagonist for FFAR2/GPR43 receptor, inhibited the anti-hyperplasic, anti-inflammatory effect of butyric acid. Further, topical application of butyric acid did not rescue the UVB induced skin damage and inflammatory level of IL-6 when FFAR2 receptor was knocked down by subcutaneous injection of FFAR2 siRNA. Overall, these results demonstrate that butyric acid interaction with the FFAR2 receptor is a crucial phenomenon during skin inflammation. Moreover, butyric acid-FFAR2 interactions could represent a central mechanism to account for the effect of probiotics on immune responses modulated by chronic UV radiation.

The skin microbiome plays an important role in preserving skin homeostasis. Thus, skin microbiome through fermentation activity might influence the skin immune system in a similar way to the gut under inflammatory disease conditions. A recent study demonstrated that topical application of antioxidants against UVB were found more effective than dietary supplementation because of the length of time needed to reach optimal concentrations in the skin [47]. In this current study, we determined that topical application of butyric acid, as a fermentation metabolite could act as a potent regulator of immune system during chronic UVB. Additionally, the dose of butyric acid is non-toxic to skin cells and thus may gain broad acceptance for not only therapeutics but also as an active ingredient in cosmetics with the ultimate aim of avoiding the strong effects from chronic UVB.

## 4. Materials and Methods

### 4.1. Ethics Statement

This research was conducted in strict accordance with an approved Institutional Animal Care and Use Committee (IACUC) protocol of the National Central University (NCU), Taiwan (NCU-106-016, 19 December 2017). ICR mice (8–9 week-old females; National Laboratory Animal Centre, Taipei, Taiwan) were sacrificed using dry ice in a closed box. 

### 4.2. Chemicals

Butyric acid (Sigma), acetonitrile, H_3_PO_4_, ether anhydrous, formaldehyde, hydrochloric acid (HCl, 37%), sodium hydroxide (NaOH), Sodium dihydrogenphosphate (NaH_2_PO_4_), and phosphoric acid (H_3_PO_4_) were purchased from J.T.Baker, Avantor Performance Materials Taiwan Co. Ltd. DMSO and glycerol were purchased from Sigma. PBS was purchased from Gibco (Gaithersburg, MD, USA). GLPG0974 (FFAR2 inhibitor) was also obtained from Tocris Bioscience (Ellisville, MO, USA). Butyric acid was obtained from Sigma (St. Louis, MO, USA).

### 4.3. Bacterial Culture

*S. epidermidis* bacteria, from ATCC 12228 was cultured on 3% tryptic soy broth (TSB) (Sigma, St. Louis, MO, USA) agar plates overnight at 37 °C. A single colony was inoculated in 3% TSB medium (Difco, Becton Dickinson UK Ltd., Oxford, UK) and cultured at 37 °C until the logarithmic growth phase. Bacterial pellets were harvested by centrifugation at 5000× *g* for 10 min, washed with PBS, and suspended in PBS.

### 4.4. Fermentation of Bacteria

*S. epidermidis* (10^7^ CFU/mL) were incubated in 10 mL rich media (10 g/L yeast extract (Biokar Diagnostics, Beauvais, France), 3 g/L TSB, 2.5 g/L K_2_HPO_4_, and 1.5 g/L KH_2_PO_4_) in the absence and presence of glycerol (2%) under aerobic conditions at 37 °C with shaking at 200 rpm. The rich media glycerol (2%) without bacteria were included as a control. The 0.002% (*w*/*v*) phenol red (Sigma) in rich media with glycerol (2%) acted as a fermentation indicator. A color change from red-orange to yellow indicated the occurrence of bacterial fermentation, which was detected as absorbance at 560 nm.

### 4.5. Inhibition of ALS Activity by Furfural

ALS (EC: 4.1. 3.18) is responsible for the conversion of pyruvate to acetolactate which can be eventually metabolized to diacetyl and 2,3-butanediol. The rate of reaction was monitored by the depletion of NADH at 340 nm during the conversion of pyruvate to 2,3-butanediol. The reaction mixture contained 70 mM sodium acetate buffer (pH 5.4), 0.17 mM thiamine pyrophosphate, and lysate (10 mg) of heat (100 °C) killed *S. epidermidis*. The reaction was started by addition of pyruvate [48] in the presence or absence of 0.4% furfural at 45 °C for 5 min. Absorbance was measured with a visible spectrophotometer at 340 nm. Values of ALS activity were expressed in enzyme as unit per mg (U/mg), in which one unit of ALS is defined as the amount of enzyme able to produce 0.1 absorbance unit per min.

### 4.6. Ultraviolet Light Exposure

ICR mice were exposed to chronic UVB with a ramping dose using an UV lamp (Model EB-280C, Spectronics Corp., Westbury, NY, USA). The UVB light source was set at a distance such that the dorsal surface of each mouse received 195 mJ/cm^2^. Dorsal skin was exposed three times per week, followed by topical application with *S. epidermidis* (10^7^) and glycerol (2%) in PBS one time per week. Whole UVB radiation and treatment was conducted for 4 weeks. Animals were photographed at 0, 16, and 30 day for measurement of wound from UVB and recovery from treatment. Animals were sacrificed 3 day following the last dose and whole skin samples, including epidermis and dermis were collected. Tissue samples (approximately 1 cm^2^) were taken from similar dorso-caudal locations for histological examination. The remaining portions of the dorsal skin were used to prepare whole skin lysate for ELISA analyses. Hematoxylin and eosin (H&E) staining was performed by the Histopathology Shared Resources Core at Li-Tzung Biotechnology Inc. Quantitative measurements of skin thickness from H&E were made using ImageJ software. 

### 4.7. Wound Measurement

To investigate the wound improvement of ICR hairless mice skin induced by UVB irradiation, the mice were transiently anesthetized using isoflurane, and the wound were measured by ImageJ software by calculating the percent of remaining wound area to the total area of skin exposed at 0, 16, and 30 day of UVB irradiation. Wounds were photographed using a USB digital microscope.

### 4.8. Histological Analysis

Histological analysis was performed to determine the epidermal thickness. Dorsal skin samples of the experimental groups were obtained after 4 weeks of UVB exposure, fixed with 10% formalin and embedded in paraffin. Sections were stained with H&E to examine epidermal thickness. Epidermal thickness was measured at 10× magnification using Olympus BX63 microscope (Olympus, Tokyo, Japan). 

### 4.9. siRNA-Mediated Gene Silencing of GPR43/FFAR2

In order to silence GPR43 gene, we used the chemically-modified siRNA that targets GPR43 receptor and the siRNA negative control, which does not target any known sequence were obtained from GenePharma Co. (Shanghai, China). Their oligonucleotide sequences are siFFAR2: sense strand, 5′-CCGGUGCAGUACAAGUUAUTT-3′; anti-sense strand, 5′-AUAACUUGUACUGCACCGGTT-3′. SiControl: sense strand, 5′-UUCUCCGAACGUGUCACGUTT; anti-sense strand, 5′-ACGUGACACGUUCGGAGAATT-3′. These chemically-modified siRNAs were delivered by intradermal injection in dorsal skin of mice using a microneedle [49,50]. 

### 4.10. Drug Treatment

Selective FFAR2 antagonist GLPG0974 (0.1 or 1 mg/kg ig) was administered to ICR mice by gavage feeding [51]. GLPG0974 was dissolved in DMSO (0.01% in saline) and DMSO (0.01% in saline) was used as the vehicle control. Butyric acid (4 mM) was topically applied on dorsal skin of mice. Butyric acid was dissolved in double distilled water.

### 4.11. ELISA

Skin samples from ICR mice were collected after 4 weeks of UVB exposure and were then lysed with T-PER™ Tissue Protein Extraction Reagent (ThermoFisher Scientific, Waltham, MA, USA) supplemented with an ethylenediaminetetraacetic acid (EDTA)-free protease inhibitor cocktail (Sigma-Aldrich, St Louis, MO, USA). Supernatant was collected from UVB irradiated and non-irradiated human keratinocytes (CCD 1106 KERTr) with and without BA treatment. IL-6 concentration in mice skin and IL-6 and IL-8 concentration in supernatant from human keratinocytes were determined by mouse or human IL-6 and IL-8 ELISA assay kit (R&D Systems, Minneapolis, MN, USA) following previous published protocol [52]. 

### 4.12. HPLC

Media samples were vortexed and equilibrated at room temperature for 5 min. Thereafter 100 µL of concentrated HCl was added per mL of each sample, followed by a vortex-mixing step of 15 s. The samples were extracted for 20 min (by gently rolling) using 5 mL of diethyl ether. After centrifugation (5 min, 3500 rpm), the supernatant was transferred to another Pyrex extraction tube and 500 µL of a 1 M solution of NaOH was added. The samples were extracted again for 20 min, followed by a centrifugation step. The aqueous phase was transferred to an auto sampler vial and 100 µL of concentrated HCl was added. After vortex mixing, 10 µL was injected onto the HPLC–UV apparatus. The mobile phase consisted of 20 mM of NaH_2_PO_4_ in HPLC water (pH adjusted to 2.2 using phosphoric acid) (A) and acetonitrile (B). The UV detector was set at a wavelength of 210 nm for detection of butyric acid in sample.

### 4.13. Statistical Analysis

Data analysis was performed by unpaired t-test or by one-way ANOVA using GraphPad Prism^®^ software. The *p*-values of <0.05 (*), <0.01 (**), and <0.001 (***) were considered significant. The mean ± standard error (SE) for at least three independent experiments was calculated. 

## Figures and Tables

**Figure 1 ijms-20-04477-f001:**
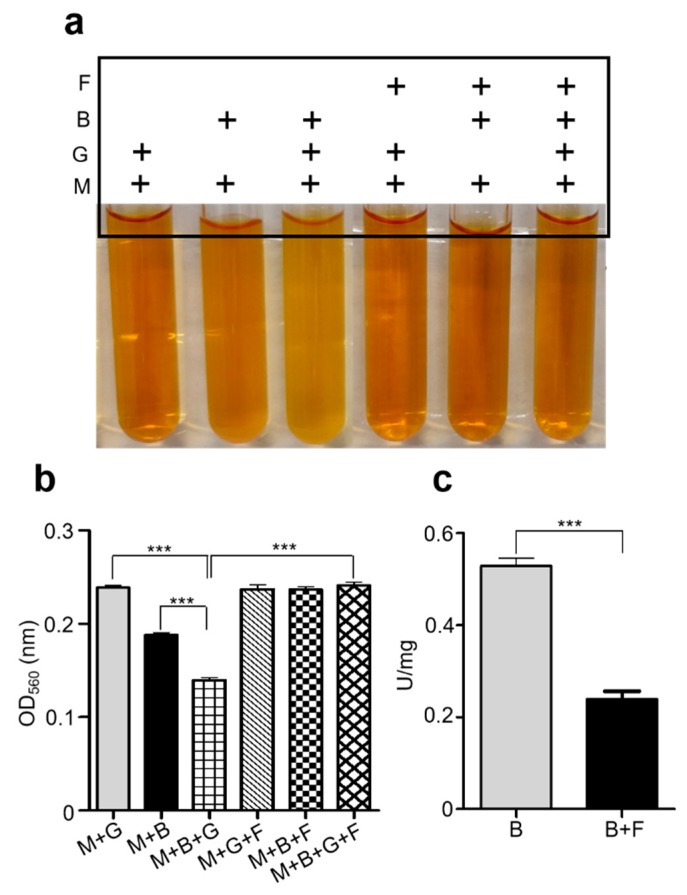
*S. epidermidis* mediates glycerol fermentation by ALS (acetolactate synthase) enzyme activity. (**a**) *S. epidermidis* bacteria (B) (10^7^ CFU/mL) were incubated in rich media (M) containing phenol red with or without 2% glycerol (G) and in the presence and absence of fermentation inhibitor furfural (F) for 12 h. Media with glycerol (M + G), bacteria (M + B), furfural (M + F), glycerol plus furfural (M + G + F), or bacteria plus furfural (M + B + F) were taken as controls. Bacterial fermentation was indicated by the color change of phenol red to yellow (arrow). (**b**) A graph showing the OD_560_ value in all the above groups. (**c**) ALS activity by furfural. The reaction mixture containing lysates of *S. epidermidis* bacteria (B) (10^7^ CFU/mL) was incubated with and without furfural (F). The activity (U/mg) of ALS in *S. epidermidis* bacteria was quantified. Data shown represent the mean ± SE of experiment performed in triplicate. *** *p* < 0.001 (two-tailed *t*-test).

**Figure 2 ijms-20-04477-f002:**
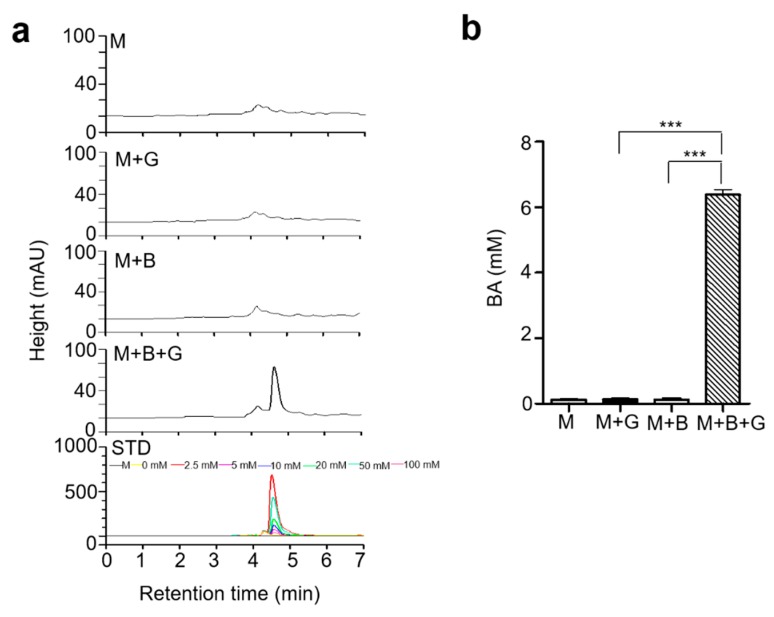
High-performance liquid chromatography-ultraviolet (HPLC-UV) analysis of the butyric acid. The fermented media from glycerol (G) fermentation by *S. epidermidis* bacteria (B) analyzed by HPLC. (**a**) Chromatograph of only media (M), media with glycerol (M + G), media with bacteria (M + B), and media with bacteria plus glycerol (M + B + G). The x-axis is retention time in minutes, and the y-axis is the milli-absorbance unit at 210 nm. (**b**) The concentrations of BA (butyric acid) was quantified from the height of butyric acid standard (STD) peaks. Data shown represent the mean ± SE of experiment performed in triplicate. *** *p* < 0.0001 (two-tailed *t*-test).

**Figure 3 ijms-20-04477-f003:**
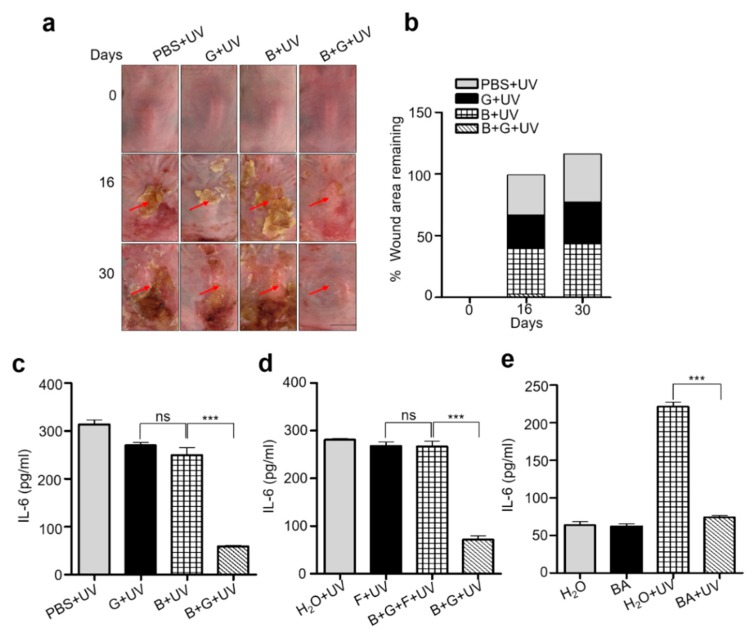
Attenuation of UVB induced inflammation by application of mixture of *S. epidermidis* and glycerol and its fermentation metabolite, butyric acid. (**a**) Skin morphology of ICR mice, topically applied with PBS, glycerol (G) (2%), *S. epidermidis* bacteria (B) (10^7^ CFU/mL), or mixture of *S. epidermidis* bacteria plus glycerol (B + G) followed by exposure with UVB (195 mJ/cm^2^) irradiation are shown at 0, 16, and 30 day. Skin lesions are indicated by red arrows. Scale bar = 5 mm. Graph (**b**) indicates percent wound area remaining of mice skin. The level of IL-6 in skin (**c**) from above all groups of ICR mice was quantified using a mouse IL-6 enzyme-linked immunosorbent assay (ELISA) kit (R&D Systems). (**d**) The IL-6 level in skin of ICR mice topically applied with H_2_O, furfural (0.4%), *S. epidermidis* bacteria (B) (10^7^ CFU/mL), glycerol (G) (2%) plus furfural (F) (0.4%), (B + G + F) or a mixture of *S. epidermidis* bacteria plus glycerol (B + G) followed by exposure with UVB (195 mJ/cm^2^) irradiation is shown. (**e**) The IL-6 level in skin of ICR mice topically applied with H_2_O and butyric acid (BA) (4 mM) followed by exposure with UVB (195 mJ/cm^2^) irradiation is shown. Data are the means of three individual experiments using five mice per group. ns = non-significant. *** *p* < 0.001. (two-tailed *t*-tests).

**Figure 4 ijms-20-04477-f004:**
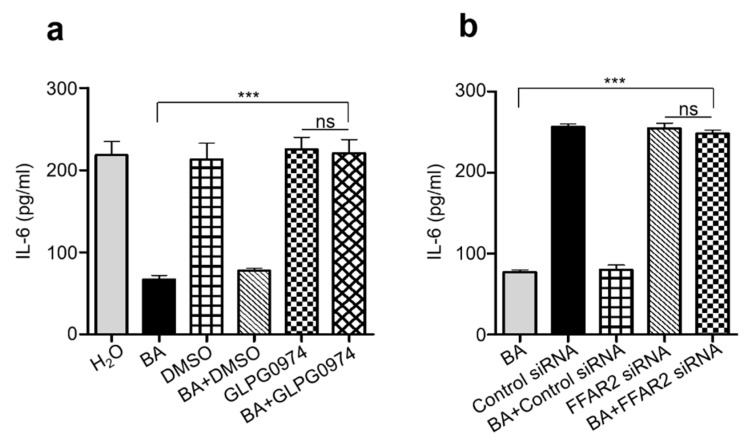
Blocking FFAR2 prevents the butyric acid mediated amelioration of inflammation. (**a**) The level of IL-6 in the dorsal skin of ICR mice upon gavage feeding of FFAR2 antagonist GLPG0974 dissolved in dimethyl sulfoxide (DMSO) in saline (0.01%) and topically applied butyric acid (BA) (4 mM) and was measured by ELISA. Mice fed with DMSO in saline and topically applied with H_2_O were included as control. (**b**) The level of IL-6 in the dorsal skin of ICR mice subcutaneously injected with FFAR2 siRNA 10 min prior to topical application with butyric acid (BA) (4 mM) was measured by ELISA. Mice injected with scrambled or negative control siRNA and topically applied with H_2_O were included as control. Data are the means of three separate experiments using five mice per group. ns = non-significant. *** *p* < 0.001. (two-tailed *t*-tests).

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
