# Peer review of "Butyric Acid from Probiotic Staphylococcus epidermidis in the Skin Microbiome Down-Regulates the Ultraviolet-Induced Pro-Inflammatory IL-6 Cytokine via Short-Chain Fatty Acid Receptor"

_ijms, 2019, doi:10.3390/ijms20184477_

Round 1

Reviewer 1 Report

The authors showed that butyrate from S. epidermidis suppressed IL-6 expression by UV-irradiated keratinocytes, resulting in amelioration of UV-induced skin damage. This is a well-written paper that can lead to clinical application of butyrate in the future. I have some minor concerns.

Results 2.1. Although the authors demonstrated that furfural inhibited both S. epidermidis-induced glycerol fermentation and ALS activity, they did not show that ALS was associated with glycerol fermentation by S. epidermidis. To conclude that “S. epidermidis induced glycerol fermentation, which required furfural-sensitive ALS activity”, experiments using ALS-deficient mutant are needed. Otherwise, the conclusion should be replaced with a weaker expression.

Figure 3b. The results of too many conditions appeared in one figure. I recommend separating this figure to two figures (figure with UVB-unexposed group and figure with UVB-exposed group). In addition, the numbers which would not be involved in this figure are contained. Please delete them.

Figure 3d did not appear in the manuscript. Probably, the term will be added in P5, L183.

Figure S2e and S2j should be exchanged each other.

It would be more interesting to investigate the effect of butyrate on human keratinocyte cell lines, such as HaCaT cells and normal human epidermal keratinocytes (NHEKs), after UVB irradiation. This experiment will strengthen the author’s conclusion that “topical application of butyric acid, as a fermentation metabolite could act as a potent regulator of immune system during chronic UVB”.

Author Response

Reviewer 1

The authors showed that butyrate from S. epidermidis suppressed IL-6 expression by UV-irradiated keratinocytes, resulting in amelioration of UV-induced skin damage. This is a well-written paper that can lead to clinical application of butyrate in the future. I have some minor concerns.

Comment 1

Results 2.1. Although the authors demonstrated that furfural inhibited both S. epidermidis-induced glycerol fermentation and ALS activity, they did not show that ALS was associated with glycerol fermentation by S. epidermidis. To conclude that “S. epidermidis induced glycerol fermentation, which required furfural-sensitive ALS activity”, experiments using ALS-deficient mutant are needed. Otherwise, the conclusion should be replaced with a weaker expression.

Response1

Here S. epidermidis uses glycerol as carbon source for fermentation, to produce pyruvate, which further converted to α-acetolactate by acetolactate synthase (ALS) and α-acetolactate converted to isobutyrate to ultimately butyrate in subsequent pathways (references cited). α-acetolactate by ALS is the major enzyme from S. epidermidis responsible for the production of butyrate (as SCFA) from glycerol as a carbon source. We have cited a new reference from our recent study demonstrating ALS activity in S. epidermidis is inhibited by furfural derivative 5-methyl furfural (5MF). However, in the future study, we could use ALS-deficient mutant S. epidermidis for the detailed analysis of the role of ALS activity for S. epidermidis fermentation.

14. Kumar, M.; Myagmardoloonjin, B.; Keshari, S.; Negari, I.P.; Huang, C.-M. 5-Methyl Furfural Reduces the Production of Malodors by Inhibiting Sodium l-Lactate Fermentation of Staphylococcus epidermidis: Implication for Deodorants Targeting the Fermenting Skin Microbiome. Microorganisms 2019, 7, 239.

34. Perrine, S.P.; Dover, G.H.; Daftari, P.; Walsh, C.T.; Jin, Y.; Mays, A.; Faller, D.V. Isobutyramide, an orally bioavailable butyrate analogue, stimulates fetal globin gene expression in vitro and in vivo. British Journal of Haematology 1994, 88, 555-561, doi:10.1111/j.1365-2141.1994.tb05073.x.

 35. Vassilev, I.; Hernandez, P.A.; Batlle-Vilanova, P.; Freguia, S.; Krömer, J.O.; Keller, J.; Ledezma, P.; Virdis, B. Microbial Electrosynthesis of Isobutyric, Butyric, Caproic Acids, and Corresponding Alcohols from Carbon Dioxide. ACS Sustainable Chemistry & Engineering 2018, 6, 8485-8493, doi:10.1021/acssuschemeng.8b00739.

Comment 2

Figure 3b. The results of too many conditions appeared in one figure. I recommend separating this figure to two figures (figure with UVB-unexposed group and figure with UVB-exposed group). In addition, the numbers which would not be involved in this figure are contained. Please delete them.

Response 2

We have separated the figure with UVB-unexposed groups/controls from figure 3b to supplementary figure 1.

Comment 3

Figure 3d did not appear in the manuscript. Probably, the term will be added in P5, L183.

 Response3

We have added the term, “Figure 3d” in the mentioned page.

Comment 4

Figure S2e and S2j should be exchanged each other.

Response 4

We have exchanged the mentioned figures

Comment 5

It would be more interesting to investigate the effect of butyrate on human keratinocyte cell lines, such as HaCaT cells and normal human epidermal keratinocytes (NHEKs), after UVB irradiation. This experiment will strengthen the author’s conclusion that “topical application of butyric acid, as a fermentation metabolite could act as a potent regulator of immune system during chronic UVB”.

Response 5

We have investigated the level of IL-6 and IL-8 by ELISA in the supernatant from UVB irradiated and non-irradiated human keratinocytes (CCD 1106 KERTr) treated with and without butyric acid (BA) after 12 h of UV exposure. The level of IL-8 was used as a positive control. The new results from this experiment has kept in Figure S2a and S2b. A new reference has also been cited for supporting this investigation.

25. Yoshizumi, M.; Nakamura, T.; Kato, M.; Ishioka, T.; Kozawa, K.; Wakamatsu, K.; Kimura, H. Release of cytokines/chemokines and cell death in UVB-irradiated human keratinocytes, HaCaT. Cell Biol Int 2008, 32, 1405-1411, doi:10.1016/j.cellbi.2008.08.011.

Reviewer 2 Report

This manuscript describes that butyric acid, a fermented product from glycerol produced by S. epidermidis inhibits UV induced IL-6 production in the skin. Authors have tried to validate their data by in vivo and in vitro system. Although this is an interesting article, there are many limitations which need to be addressed-

Page 3, fig 1, authors describe almost the complete result in the legend, which is not required as they already described it in the results section. 1(b), it seems that there is a significant decrease in the OD value in (M+B) condition compared to (M+G), please explain although they did not add significance symbol here. In most cases, the authors did not add "n" value in the figure legend. Please add n values in all the figure legends. Page 5, figure-3 (a)-this panel contains too many images, they used many control conditions, that is why every image looks very small and unclear resolution. Authors could use one control condition may be B+G, and they could make the individual image larger with good resolution. They could use a separate supplemental figure with all the controls. Fig 1(b), again adding all the controls makes the figure very unclear. Symbols of all the controls with B+G+UV overlap each other in all the time point, please avoid some control in the main figure and add them in the supplementary figure. Authors could also make the bar plot instead of the dot plot here so that individual plot could be identified clearly. Protocol for the measurement of IL-6 from mouse skin is not clear, please elaborate the protocol in the method section. If they used any previously reported protocol, please add the citation with a brief description. Also, they measure IL-6 in the skin, they should see this cytokine level in the serum also. This manuscript only described the protein expression, authors should add RNA expression to validate the protein expression. There are many cytokines (IL-1, IL-8 etc.) are reported to be induced by UV, authors could use at least one as a positive control including IL-6. Their previous paper (Wang et al, 2014) reported 6 days for the fermentation process whereas this manuscript used only 12 hours, please explain. Page-10, siRNA delivery protocol is very brief. Please elaborate on the protocol in the method section. If authors used any previously published protocol, please add that citation after a brief description.

Author Response

Reviewer 2

This manuscript describes that butyric acid, a fermented product from glycerol produced by S. epidermidis inhibits UV induced IL-6 production in the skin. Authors have tried to validate their data by in vivo and in vitro system. Although this is an interesting article, there are many limitations which need to be addressed-

Comment 1

Page 3, fig 1, authors describe almost the complete result in the legend, which is not required as they already described it in the results section.

Response 1

We have précised the figure legend in page 3, fig1.

Comment 2

1(b), it seems that there is a significant decrease in the OD value in (M+B) condition compared to (M+G), please explain although they did not add significance symbol here.

Response 2

Here we have used Rich media, which contain few carbon metabolites that might have used by S. epidermidis bacteria as a carbon source during bacteria growth turning phenol red color from red to yellowish-orange and the OD value decreased compared to media only. However, in the medium with glycerol and S. epidermidis bacteria, we have specifically used glycerol as a carbon source to detect its fermentation activity and we have detected significantly higher fermentation with glycerol changing the color of phenol red from red to yellow. We have also cited two references (no. 13 and no 15) evidencing glycerol is a carbon source for fermentation.

13. Wang, Y.; Kuo, S.; Shu, M.; Yu, J.; Huang, S.; Dai, A.; Two, A.; Gallo, R.L.; Huang, C.M. Staphylococcus epidermidis in the human skin microbiome mediates fermentation to inhibit the growth of Propionibacterium acnes: implications of probiotics in acne vulgaris. Appl Microbiol Biotechnol 2014, 98, 411-424, doi:10.1007/s00253-013-5394-8.

15. Yang, A.J.; Marito, S.; Yang, J.J.; Keshari, S.; Chew, C.H.; Chen, C.C.; Huang, C.M. A Microtube Array Membrane (MTAM) Encapsulated Live Fermenting Staphylococcus epidermidis as a Skin Probiotic Patch against Cutibacterium acnes. Int J Mol Sci 2018, 20, doi:10.3390/ijms20010014.

Comment 3

In most cases, the authors did not add "n" value in the figure legend. Please add n values in all the figure legends

Response 3

We have added "n" value in all the figure legend.

Comment 4

Page 5, figure-3 (a)-this panel contains too many images, they used many control conditions, that is why every image looks very small and unclear resolution. Authors could use one control condition may be B+G, and they could make the individual image larger with good resolution. They could use a separate supplemental figure with all the controls.

Response 4

We have moved the UVB-unexposed groups/ controls (PBS, G, B, and BG) from Figure 3a, 3b and 3c to a separate supplementary figure 1.

Comment 5

Fig 3(b), again adding all the controls makes the figure very unclear. Symbols of all the controls with B+G+UV overlap each other in all the time point, please avoid some control in the main figure and add them in the supplementary figure. Authors could also make the bar plot instead of the dot plot here so that individual plot could be identified clearly.

Response 5

In Figure 3b, we have shown figure only with UVB-exposed group and we have moved UVB-unexposed groups/controls to a separate supplementary figure 1. We have changed the dot plot graphs to bar graphs for the percent wound area remaining from all experiments.

Comment 6

Protocol for the measurement of IL-6 from mouse skin is not clear, please elaborate the protocol in the method section. If they used any previously reported protocol, please add the citation with a brief description. Also, they measure IL-6 in the skin, they should see this cytokine level in the serum also.

Response 6

 We have elaborated the protocol for the measurement of IL-6 from mouse skin and cited a new reference for this method.

 52. Keshari, S.; Sipayung, A.D.; Hsieh, C.C.; Su, L.J.; Chiang, Y.R.; Chang, H.C.; Yang, W.C.; Chuang, T.H.; Chen, C.L.; Huang, C.M. The IL-6/p-BTK/p-ERK signaling mediates the calcium phosphate-induced pruritus. FASEB J 2019, 10.1096/fj.201900016RR, fj201900016RR, doi:10.1096/fj.201900016RR.

In our future study, we will measure the level of IL-6 in the in the serum of UVB exposed mice.

Comment 7

This manuscript only described the protein expression; authors should add RNA expression to validate the protein expression. There are many cytokines (IL-1, IL-8 etc.) are reported to be induced by UV, authors could use at least one as a positive control including IL-6.

Response 7

 We have detected mRNA expression of FFAR2 by RTPCR analysis to validate the FFAR2 knockdown by FFAR2 antagonist, GLPG0974 and chemically modified-siRNA delivery. The new FFAR2 mRNA expression data has been included in  Figure S4f and s4l.

 We have investigated the level of IL-6 and IL-8 by ELISA in the supernatant from UVB irradiated and non-irradiated human keratinocytes (CCD 1106 KERTr) treated with and without butyric acid (BA) after 12 h of UV exposure, where the level of IL-8 was served as a positive control. The new results from this experiment were kept in Figure S2a and S2b.

 A new reference has also been cited for supporting this investigation.

 25. Yoshizumi, M.; Nakamura, T.; Kato, M.; Ishioka, T.; Kozawa, K.; Wakamatsu, K.; Kimura, H. Release of cytokines/chemokines and cell death in UVB-irradiated human keratinocytes, HaCaT. Cell Biol Int 2008, 32, 1405-1411, doi:10.1016/j.cellbi.2008.08.011.

Comment 8

 Their previous paper (Wang et al, 2014) reported 6 days for the fermentation process whereas this manuscript used only 12 hours, please explain.

Response 8

 In the previous paper (Wang et al, 2014), human skin containing a mixture of microorganisms were grown in TSB agar and S. epidermdis were identified by 16S rRNA gene sequencing from the colony of skin microorganisms that created an inhibition zone within a P. acnes colony. However, in the current study, we have directly used S. epidermdis (ATCC12228) and checked fermentation for 12 h by following the recent studies that we have cited in reference no. 14 and 15 in the manuscript.

 14. Kumar, M.; Myagmardoloonjin, B.; Keshari, S.; Negari, I.P.; Huang, C.-M. 5-Methyl Furfural Reduces the Production of Malodors by Inhibiting Sodium l-Lactate Fermentation of Staphylococcus epidermidis: Implication for Deodorants Targeting the Fermenting Skin Microbiome. Microorganisms 2019, 7, 239.

 15. Yang, A.J.; Marito, S.; Yang, J.J.; Keshari, S.; Chew, C.H.; Chen, C.C.; Huang, C.M. A Microtube Array Membrane (MTAM) Encapsulated Live Fermenting Staphylococcus epidermidis as a Skin Probiotic Patch against Cutibacterium acnes. Int J Mol Sci 2018, 20, doi:10.3390/ijms20010014.

Comment 9

Page-10, siRNA delivery protocol is very brief. Please elaborate on the protocol in the method section. If authors used any previously published protocol, please add that citation after a brief description.

Response 9

 We have elaborated the siRNA delivery protocol in the method section and cited two references for the delivery method.

 49. Pan, J.; Ruan, W.; Qin, M.; Long, Y.; Wan, T.; Yu, K.; Zhai, Y.; Wu, C.; Xu, Y. Intradermal delivery of STAT3 siRNA to treat melanoma via dissolving microneedles. Scientific Reports 2018, 8, 1117, doi:10.1038/s41598-018-19463-2.

50. Dou, S.; Yao, Y.-D.; Yang, X.-Z.; Sun, T.-M.; Mao, C.-Q.; Song, E.-W.; Wang, J. Anti-Her2 single-chain antibody mediated DNMTs-siRNA delivery for targeted breast cancer therapy. Journal of Controlled Release 2012, 161, 875-883, doi:https://doi.org/10.1016/j.jconrel.2012.05.015.

Reviewer 3 Report

This is a well written manuscript describing novel work investigatig the effects of the skin commensal, S. epidermidis, on the response of skin to ultraviolet radiation.  The study is novel and timely and should be if great interest to others in the field. I have two comments:

1) The authors show fermentation of glycerol by S. epidermidis and butyrate production as a product. PLease could they comment somewhere in the article, as to the abundance of glycerol on skin. What would eb the likely physiological relevece of this finding to human skin?

2) In figure 2 part A - what is 'mAU'on the Y axis? I assume arbitary units but please specify. ON figure part B, what does BA stand for?

Author Response

Reviewer 3

This is a well-written manuscript describing novel work investigating the effects of the skin commensal, S. epidermidis, on the response of skin to ultraviolet radiation.  The study is novel and timely and should be if great interest to others in the field. I have two comments:

Comment1

The authors show fermentation of glycerol by S. epidermidis and butyrate production as a product. Please could they comment somewhere in the article, as to the abundance of glycerol on skin. What would be the likely physiological relevance of this finding to human skin?

Response 1

 We have added new information regarding the abundance of glycerol on the skin and physiological relevance of our finding to human skin in the discussion part.

 Glycerol is a major component in stratum corneum (SC) hydration with the varied amount in different parts of human skin (0.7 ug cm-2 in the cheek and  0.l ug cm-2 in the forearm and sole). Multiple diseases characterized by xerosis and impaired epidermal barrier function, such as atopic dermatitis were improved by topical application of glycerol. Although glycerol could not penetrate deeper in SC but the short chain fatty acids (SCFAs)  generated from glycerol fermentation by skin commensal bacteria, may modulate physiological processes such as skin barrier homeostasis and inflammation. Three references have been cited.

Yang, A.J.; Marito, S.; Yang, J.J.; Keshari, S.; Chew, C.H.; Chen, C.C.; Huang, C.M. A Microtube Array Membrane (MTAM) Encapsulated Live Fermenting Staphylococcus epidermidis as a Skin Probiotic Patch against Cutibacterium acnes. Int J Mol Sci 2018, 20, doi:10.3390/ijms20010014.

Fluhr, J.W.; Darlenski, R.; Surber, C. Glycerol and the skin: holistic approach to its origin and functions. Br J Dermatol 2008, 159, 23-34, doi:10.1111/j.1365-2133.2008.08643.x.

Lin, T.K.; Zhong, L.; Santiago, J.L. Anti-Inflammatory and Skin Barrier Repair Effects of Topical Application of Some Plant Oils. Int J Mol Sci 2017, 19, doi:10.3390/ijms19010070.

Comment2

In figure 2 part A - what is 'mAU'on the Y axis? I assume arbitary units but please specify. ON figure part B, what does BA stand for?

Response2

 In figure 2 part A - 'mAU' on the Y-axis stands for milli-Absorbance units (number of absorption of analytes).  We have added the mAU abbreviation in the manuscript in the figure legend part of figure 2 as follow

The x-axis is retentions time in minutes, and the y-axis is the milli absorbance unit at 210 nm.

On figure part B, BA stands for butyric acid. We have added the BA abbreviation in the manuscript in the figure legend part of figure 2.

Round 2

Reviewer 1 Report

I have no more concerns.

Reviewer 2 Report

Responses to all comments are satisfactory.